# Vitrification and nanowarming enable long-term organ cryopreservation and life-sustaining kidney transplantation in a rat model

Zonghu Han [1,7], Joseph Sushil Rao [2,3,7], Lakshya Gangwar[1], Bat-Erdene Namsrai[2], Jacqueline L. Pasek-Allen [1,4], Michael L. Etheridge [1], Susan M. Wolf [5], Timothy L. Pruett [2], John C. Bischof [1,4,6,8] & Erik B. Finger [2,8]

Banking cryopreserved organs could transform transplantation into a planned procedure that more equitably reaches patients regardless of geographical and time constraints. Previous organ cryopreservation attempts have failed primarily due to ice formation, but a promising alternative is vitrification, or the rapid cooling of organs to a stable, ice-free, glass-like state. However, rewarming of vitrified organs can similarly fail due to ice crystallization if rewarming is too slow or cracking from thermal stress if rewarming is not uniform. Here we use "nanowarming," which employs alternating magnetic fields to heat nanoparticles within the organ vasculature, to achieve both rapid and uniform warming, after which the nanoparticles are removed by perfusion. We show that vitrified kidneys can be cryogenically stored (up to 100 days) and successfully recovered by nanowarming to allow transplantation and restore life-sustaining full renal function in nephrectomized recipients in a male rat model. Scaling this technology may one day enable organ banking for improved transplantation.

The ability to intentionally and reproducibly cryopreserve living biological systems, including cells, tissues, organs, and even whole organisms, began in 1949 with the preservation of fowl sperm using glycerol, a cryoprotective agent (CPA) that protected the sperm cells during freezing[1]. That work was followed by important proof-of-principle cryopreservation of mammalian blood and embryos with other CPAs[2,3]. These and other studies also demonstrated that injury from ice crystallization during freezing limited success, especially in larger systems[4–9]. Efforts to address this barrier led to "vitrification," an

approach using higher concentrations of CPAs and faster rates of cooling that avoided crystallization entirely by forming a glassy state during cooling, as demonstrated in both embryos[10] and even whole rabbit kidneys[11]. By avoiding crystallization during both cooling and rewarming, mammalian embryo cryopreservation became a reality and transformed the field of reproductive technology. However, preventing crystallization during rewarming in larger bulk systems, like whole kidneys, remains elusive due to the inability of conventional convective rewarming (i.e., surface warming) to provide rapid and

[1]Department of Mechanical Engineering, University of Minnesota, Minneapolis, MN, USA. [2]Department of Surgery, University of Minnesota, Minneapolis, MN, USA. [3]Schulze Diabetes Institute, University of Minnesota, Minneapolis, MN, USA. [4]Department of Biomedical Engineering, University of Minnesota, Minneapolis, MN, USA. [5]Consortium on Law and Values in Health, Environment & the Life Sciences, University of Minnesota, Minneapolis, MN, USA. [6]Institute for Engineering in Medicine, University of Minnesota, Minneapolis, MN, USA. [7]These authors contributed equally: Zonghu Han and Joseph Sushil Rao. [8]These authors jointly supervised this work: John C. Bischof, Erik B. Finger. ✉e-mail: bischof@umn.edu; efinger@umn.edu

uniform heating rates across these larger scales. Indeed, vitrification followed by long-term transplant success with a kidney (or any organ) has never been reproducibly achieved.

Nevertheless, interest in organ cryopreservation has remained high due the short preservation limits for kidneys and other organs. Organ banking via cryopreservation would make organ transplantation an elective rather than an urgent/emergent procedure and revolutionize how organs are used to treat human disease. Better donor/recipient matching, improved equity in access, better patient preparation, better transplant tolerance protocols, increased organ utilization, and enhanced graft and patient survival could all be enabled by long-term organ banking.

The critical warming rates (CWR) needed to avoid ice crystallization during rewarming are typically an order of magnitude higher (10–1000 s °C/min) than the required critical cooling rates (CCR) (1–100 s °C/min), even with the aid of CPAs[12,13]. Further, temperature non-uniformity during rewarming produces thermal stress that can cause cracking. Thus, speed and uniformity of rewarming are the primary obstacles to success with vitrification.

To address these obstacles, we have developed "nanowarming," which achieves both objectives simultaneously by generating heat from within and throughout the organ rather than just at its surface[14]. In nanowarming, iron oxide nanoparticles (IONPs) are perfused throughout the organ vasculature along with CPA solutions. The organ is vitrified and then rewarmed on-demand by placing it in a radiofrequency (RF) coil that induces alternating magnetic fields from electric current flowing through the coil. The magnetic fields then generate an oscillatory response in the nanoparticles that generate heat throughout the system. Notably, nanowarming rates are not dependent on system size or boundary conditions since the RF frequencies used penetrate tissues without attenuation[14–16]. In addition, perfusion within the capillaries allows sufficiently uniform delivery of CPAs and IONPs regardless of organ size. Thus, nanowarming is intrinsically scalable to human-sized organs for clinical translation. Importantly, since the IONPs remain in the organ vasculature, they can be washed out during CPA perfusion unloading.

In the past, we and others have shown that nanowarming can rewarm vitrified organs (including kidneys) from animal models with physical success, but only partial biological recovery and no transplant data[17–20]. We found that CPA damage, not physical injury from vitrification and nanowarming, was the limiting step for biologic and functional recovery. We hypothesized that if we could overcome CPA injury, nanowarming would enable the recovery of viable and functional organs following vitrification.

Here we demonstrate, for the first time, both the physical and biological success of vitrifying and nanowarming an organ. By applying engineering principles, we optimized the CPA loading for low toxicity vitrification. That, combined with rapid and uniform nanowarming, enabled cryopreservation of kidneys for 1–100 days and on-demand rewarming. Recovery of organ function was demonstrated in vitro by normothermic machine perfusion and in vivo in a rat transplant model where both native kidneys were removed at the time of transplant. Full renal function was restored after nanowarming and transplantation, with the transplanted organs sustaining the life of the recipient animals.

## Results
### Overview of nanowarming
An overview of the kidney vitrification and nanowarming process is shown in Fig. 1. After recovery from the donor, the kidney is flushed with cold University of Wisconsin (UW) preservation solution, similar to clinical transplant practice, and connected to a pressure- and flow-controllable multithermic perfusion system. CPA is added to a carrier solution (LM5[21]) and then loaded by vascular perfusion. Loading concentration gradually increases to avoid osmotic stress due to the

hypertonicity of these solutions. The duration of CPA loading balances the time needed for CPA transport from the vasculature to the extravascular (interstitial and cellular) space against CPA toxicity, which increases with longer exposure time. IONPs are perfused along with the final step of CPA loading. The IONPs used are silica and polyethylene glycol (PEG) coated, which increases stability in CPAs and provides for biocompatibility and organ washout[18,22]. Following CPA and IONP loading, the organ is placed in a controlled rate freezer and cooled at a rate faster than the CPA's CCR to below the glass transition temperature ($T_g$, ~ −128 °C), where the system enters a stable glassy state. The vitrified organ is then transferred to a −150 °C freezer for storage. When needed, the organ is removed from cryogenic storage, placed in an RF coil, and rewarmed at a rate exceeding the CPA's CWR. The CPA and IONPs are gradually unloaded, and the organ is ready for transplantation.

### CPA loading and kidney vitrification
Our previous attempts at vitrification and nanowarming of rat kidneys were limited by damage from the CPA used (VS55)[18]. Here we changed to VMP as the CPA due to its reduced renal toxicity[21]. We used mass transport modeling to optimize the loading conditions for improved tissue CPA concentration for vitrification (see Supplementary Data and Supplementary Discussion). Figure 2a shows an example of arterial pressure, flow, and temperature during VMP perfusion with the modified protocol. We loaded IONPs at a concentration of 10 mg iron (Fe)/mL in VMP during the final 4.5 min at a constant flow of 0.5 mL/min, with a pressure of 44.9 ± 10.3 mmHg.

VMP-only treated kidneys (loading and unloading only but no nanoparticles or vitrification) were histologically similar to fresh control kidneys and much better than those treated with the prior CPA, VS55 (Fig. 3). VMP-only treated kidneys had normal glomeruli, Bowman's space, basement membranes, proximal convoluted tubules, distal convoluted tubules, collecting ducts, and vasculature. In contrast, VS55-only treated kidneys demonstrated increased Bowman's space, diffuse tubular necrosis, and hyaline changes (but with no vascular compromise). The histologic appearance confirmed that VMP

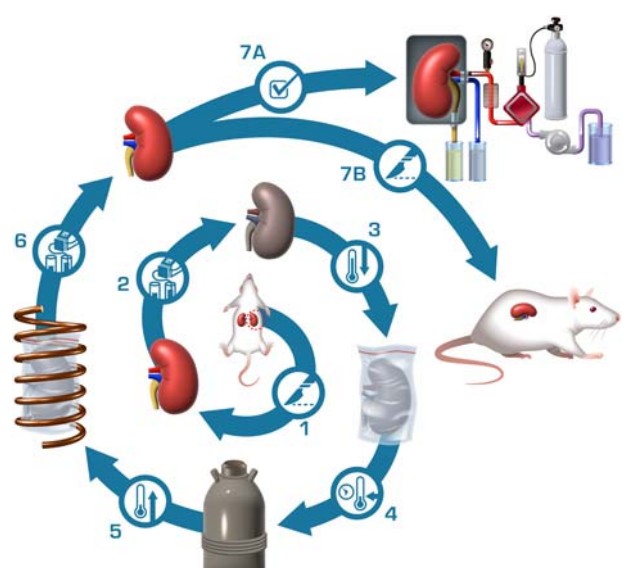

**Fig. 1 | Overview of nanowarming.** Depicted are the sequential steps of the nanowarming procedure as follows: (1) Kidney recovery from the donor; (2) Loading of cryoprotective agents (CPAs) and iron oxide nanoparticles (IONPs); (3) Rapid cooling to a vitrified state; (4) Storage at −150 °C until needed; (5) Rapid and uniform rewarming in radiofrequency (RF) alternating electromagnetic field; (6) Unloading of CPA and IONPs; and finally, organs are ready for (7A) assessment by normothermic machine perfusion or (7B) transplantation.

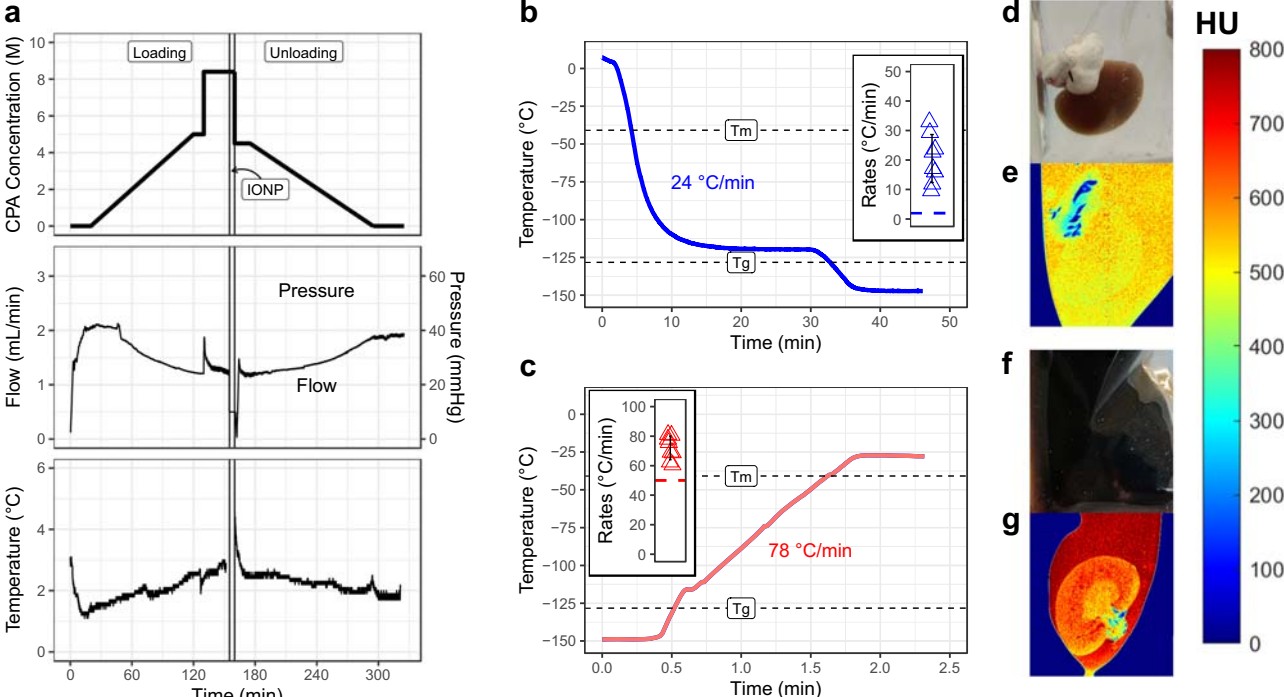

**Fig. 2 | Vitrification and nanowarming of kidneys. a** CPA concentration, perfusion pressure, arterial flow rate, and temperature profile during the loading and subsequent unloading of CPA and IONPs in rat kidneys. **b** Representative thermal profile during cooling from above the melting temperature ($T_m = -40.8\,°C$) to an annealing step just above the glass transition temperature ($T_g = -128.3\,°C$), and finally slower cooling into the glassy zone. The blue shaded area is the zone of risk for ice formation. Inset are cooling rates and the critical cooling rate (blue dashed line, CCR = 2 °C). $n = 8$. **c** Representative thermal history during nanowarming, rewarming rates, and the critical warming rate (red dashed line, CWR = 50 °C). $n = 8$. Representative photograph (**d**) and pseudocolor image acquired by micro-CT (**e**) of the vitrified VMP-loaded kidney. Representative photo (**f**) and pseudocolor image acquired by micro-CT (**g**) of the vitrified VMP + IONP-loaded kidney. The pseudocolor scale shows micro-CT image radiodensity (in Hounsfield units). Data are mean ± s.d. Source data are provided as a Source Data file. HU Hounsfield units; $T_g$ glass transition temperature; $T_m$, melting temperature.

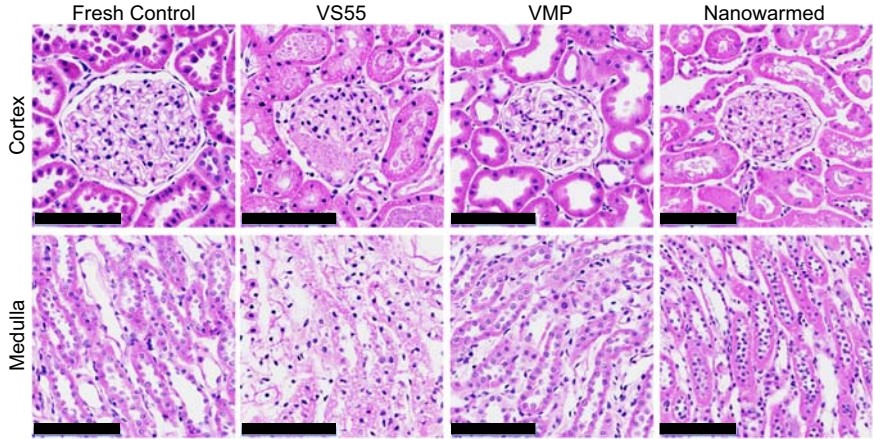

**Fig. 3 | Histologic appearance of CPA-only treated and CPA plus nanowarmed kidneys.** Histological changes (H&E) of renal cortex and medulla following treatment with CPA (VS55-only and VMP-only) and vitrification and nanowarming (N–W) compared to fresh untreated controls. Scale bars are 100 μm. Data are representative images from $n = 6$ (Fresh Control and Nanowarmed), $n = 3$ (VS55), and $n = 4$ (VMP) separate experiments for each condition. H&E, hematoxylin and eosin.

was less toxic than VS55, further assessed through ex vivo normothermic perfusion as discussed below. Further histologic details are in the Supplementary Information.

After CPA and IONP loading, the kidney was placed in a polyethylene bag containing 4 mg Fe/mL IONPs in full-strength VMP (8.4 M). A fiberoptic temperature probe was placed in the solution adjacent to the kidney to record the thermal history, and the organ was cooled to −150 °C in a controlled rate freezer and moved to a −150 °C

freezer for storage. The mean cooling rate was $20.5 \pm 8.1\,°C/min$ (Fig. 2b), which was much faster than the 2 °C/min CCR of VMP in kidney tissue.

Successful vitrification was determined by direct visual inspection and micro-computed tomography (micro-CT). Because the opaque nanoparticles obscured direct visualization, some kidneys were vitrified without IONPs (Fig. 2d, e). The surrounding CPA was translucent and glassy in appearance, and the kidney lacked the changes seen

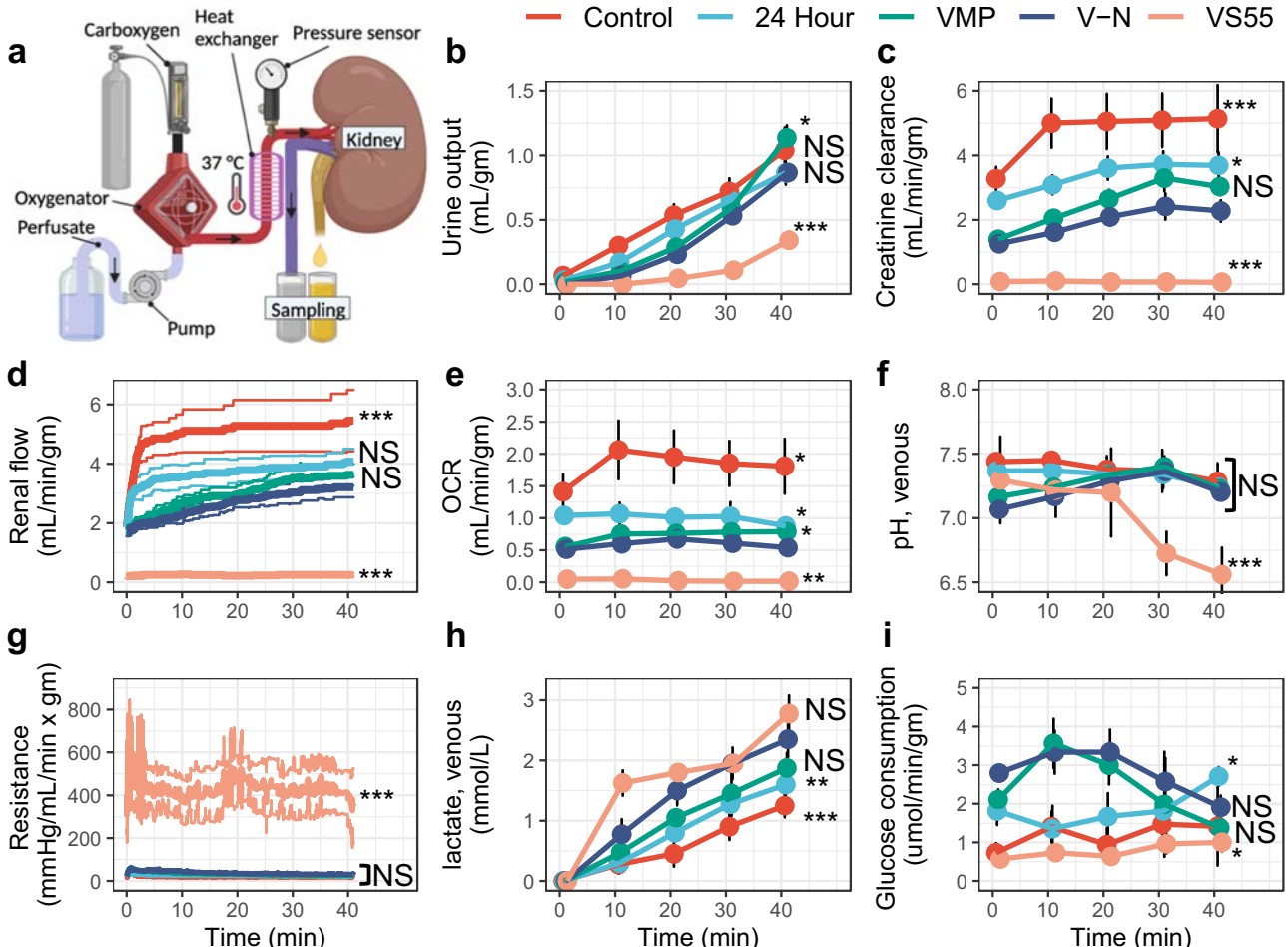

**Fig. 4 | Normothermic machine perfusion of rat kidneys using acellular blood substitute. a** Schematic of the normothermic perfusion system for diagnostic assessment of fresh untreated (control), 24-h UW preserved (24 h), VMP loaded-unloaded (VMP), VS55 loaded-unloaded (VS55), and vitrified and nanowarmed (V−N) rat kidneys. **b** Cumulative urine output over the course of perfusion. **c** Creatinine clearance. **d** Alterations in renal artery flow. **e** Oxygen consumption rate. **f** Venous pH. **g** Vascular resistance. **h** Lactate production. **i** Glucose consumption. Data are mean ± s.d. $n = 4$ per condition. Statistical comparison between nanowarmed kidney and other treatment groups at minute 40 using ANOVA and Tukey HSD (Games–Howell if unequal variance) post hoc tests. $*P < 0.05$, $**P < 0.01$, $***P < 0.001$; NS, not significant. Full statistical treatment in Supplementary materials. Source data are provided as a Source Data file. OCR oxygen consumption rate; V−N, vitrified and nanowarmed.

visually with freezing (color change to pale pink), indicating that the kidney and surrounding CPA appeared well vitrified. Some ice formed in the extrarenal adherent fat in the perihilar area, presumably due to hypoperfusion of that tissue and decreased CPA concentration. Micro-CT can discriminate between vitrified and frozen tissues based on radiodensity (Fig. 2e and Supplementary Data)[18,23]. The kidney extravascular space had a mean radiodensity of $507 \pm 30$ Hounsfield Units (HU), while the adherent fat outside the kidney was $267 \pm 104$ HU. These are consistent with successful vitrification of the kidney itself and ice present in the extrarenal fat.

For comparison, we also show a vitrified IONP-loaded kidney (Fig. 2f) where the opacity of IONPs obscured the visual appearance. Micro-CT showed successful vitrification of the CPA plus IONP-loaded kidneys, but the absolute radiodensity of vitrified vs. frozen tissue was shifted because of the IONPs (Fig. 2g). The solution, cortex, outer medulla, and inner medulla had mean HU values of $768 \pm 20$, $628 \pm 15$, $656 \pm 20$, and $560 \pm 18$, respectively. It is expected that these differences are due to regional variance in vascular density[21], leading to localized differences in IONP concentration as previously observed[18]. However, there were no areas of low HU attenuation that would suggest ice crystallization or linear transitions suggesting cracking (see Supplementary Data for micro-CT calibration of vitrified materials).

After vitrification, the kidneys were stored for 1–100 days at −150 °C.

**Kidney nanowarming and post-nanowarming assessment**
Kidney nanowarming was performed using a 15 kW RF coil at 63 kA/m and 180 kHz[14]. The measured warming rate adjacent to the kidney was $72.0 \pm 8.0$ °C/min (Fig. 2c), which is faster than the ~50 °C/min CWR of VMP in kidney tissue (Supplementary Methods).

Following nanowarming, CPA was unloaded by ramping from 4.2 M VMP (50% of full strength) plus 300 mM mannitol (to maintain osmotic balance) to 0 M CPA (LM5 carrier only) over 120 min (Fig. 2a). Venous and ureteral effluent CPA concentration at the end of unloading showed near complete CPA washout (Supplementary Data Fig. 2b).

Histologically, nanowarmed kidneys appeared similar to VMP-only treated ones and much better than VS55-only (Fig. 3). No extra "white space" was seen within the extravascular space of the nano-warmed kidneys, which would have been seen if ice had formed during the process[14].

To assess the function of nanowarmed kidneys, we utilized normothermic machine perfusion (NMP) of the kidneys with an oxygenated acellular blood substitute (Krebs–Henseleit Buffer) (Fig. 4a). During 40 min of NMP at 37 °C, hemodynamic parameters were

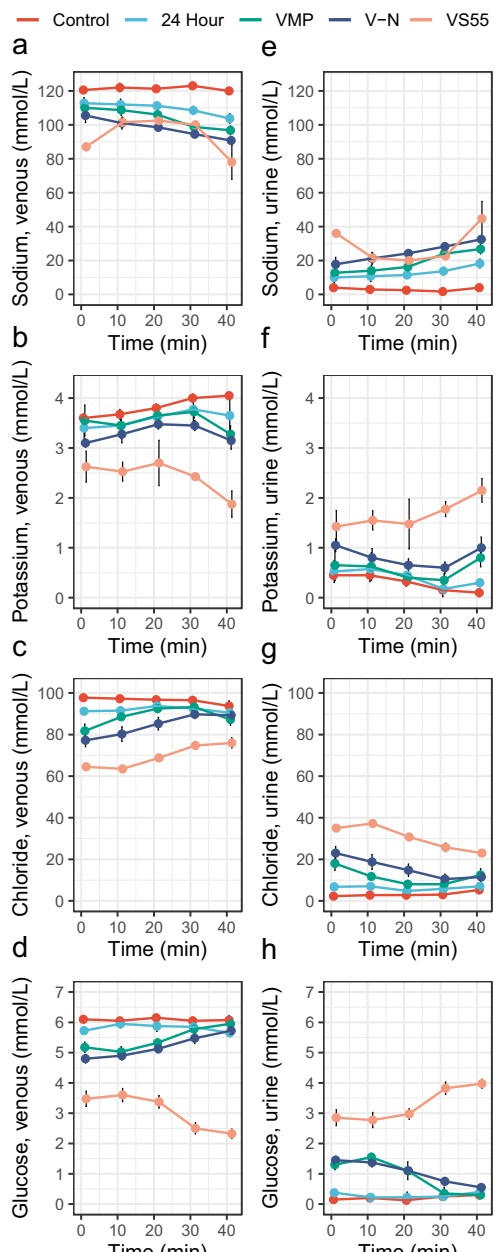

**Fig. 5 | Electrolytes and glucose during normothermic perfusion of kidneys.**
**a**–**d** Venous effluent (left) and **e**–**h** Urine (right) effluent analysis of electrolytes and glucose during 40 min of normothermic machine perfusion of fresh control, 24-h cold stored, VMP-only, and VS55-only treated kidneys were compared to nanowarmed kidneys. Measured parameters include sodium (**a**, **e**), potassium (**b**, **f**), chloride (**c**, **g**), and glucose (**d**, **h**). Data are mean ± s.d. for *n* = 4 for each group and time point. Source data are provided as a Source Data file. V–N, vitrified and nanowarmed.

recorded, and the venous and ureter effluents were sampled. We compared nanowarmed kidneys to 4 other experimental groups: fresh control, 24-h cold stored (in UW solution at 4 °C), VMP-only (CPA load and unload without IONPs, vitrification, or nanowarming), and VS55-only. For each measure tested, the trend from best to worst was fresh control; followed by 24-h cold stored, VMP-only, and nanowarmed (all with statistically similar results); and then VS55-only, which performed the worst (Fig. 4).

Nanowarmed kidneys made urine immediately upon perfusion (Fig. 4b) and at rates that were not statistically different than fresh control or cold-stored kidneys and very similar to VMP-only treated

organs, but much better than VS55-only. Creatinine clearance, a measure of overall renal function, was slightly reduced compared to fresh control and cold-stored kidneys, but similar to VMP-only (Fig. 4c). Nanowarmed kidney urine and venous electrolytes (Na+, K+, Cl−) were very similar to fresh control (Fig. 5). Urine glucose in nanowarmed kidneys normalized to <1 mmol/L, demonstrating an absence of glycosuria (Fig. 5 h).

Flow rate and vascular resistance at physiologic perfusion pressure (90–110 mmHg) provided an overall diagnostic assessment of the organs tested (Fig. 4d, g), as low vascular resistance during machine perfusion is associated with better post-transplantation performance[24]. The mean vascular resistance of nanowarmed kidneys after 40 min of perfusion was 31.3 ± 3.6 mmHg/mL/min × g, which was slightly higher but not statistically different than fresh control (18.8 ± 4.1, *P* = 0.9), 24-h cold stored (24.9 ± 2.6, *P* = 1.0) or VMP-only (27.5 ± 3.4, *P* = 1.0), but much lower than VS55-only (372 ± 131, *P* < 0.001).

Additionally, nanowarmed organs consumed oxygen and glucose during machine perfusion (Fig. 4e, i), indicating that they were metabolically active. Oxygen consumption by nanowarmed kidneys was slightly lower than fresh control but was very similar to 24-h cold stored or VMP-only. Initially, the venous pH of nanowarmed kidneys was somewhat acidic, suggesting anaerobic metabolism, but the pH normalized by 20 min of NMP (Fig. 4f). This likely reflected resumption of mitochondrial function and aerobic metabolism. Venous lactate, which correlates with organ quality in perfused organ systems[25,26], rose slowly for all groups, but the differences between groups were minor (Fig. 4h).

### Transplantation of nanowarmed kidneys
As a final measure of organ function, we tested nanowarmed kidneys in a rat transplant model where the recipient's native kidneys were removed at the time of transplant. Thus, renal function and survival of the animal depended solely on the transplanted organ. We compared outcomes between fresh control kidney transplants and nanowarmed kidneys that had been vitrified for 1–100 days prior to rewarming and transplant. Intraoperatively, all nanowarmed kidneys reperfused rapidly and homogeneously upon restoration of blood flow and appeared similar to fresh control organ transplants (Fig. 6). Nanowarmed kidneys made urine within 40–45 min following reperfusion. In contrast, fresh control transplants made urine within a few minutes. We did not observe any vascular thrombosis.

Postoperatively, all fresh control and nanowarmed kidney transplants continued to produce urine, and all animals survived for the full 30-day study period. In syngeneic (Lewis to Lewis) nanowarmed kidney recipients, serum creatinine levels (a principal measure of renal function) were higher on postop day 1 than in the control transplants (Fig. 7b). Creatinine in the nanowarmed kidney recipients continued to rise, peaking between days 2–3, and then gradually declined to reach control levels over 2–3 weeks. From day 14 onward, the creatinine in nanowarmed recipients was not statistically different from that in control kidney recipients. The creatinine fell below 2.0 mg/dL by day 19 and into the normal range for healthy rats on day 23, remaining between 0.4 and 0.8 mg/dL until the end of follow-up.

During the first two postoperative weeks, nanowarmed kidney recipients also experienced more metabolic dysfunction than control transplants. Hyperkalemia peaked on days 2–3 and slowly declined after that (Fig. 7c). Partially compensated metabolic acidosis (low pH, low HCO3-, and low pCO2) was also resolved by day 15 (Fig. 7d–f). Serum lactate levels were slightly above the normal range but normalized by days 7-10 (Fig. 7g).

Following transplant, both control and nanowarmed kidney recipients increased body weight. Initially, nanowarmed organ recipients experienced greater weight gain, presumably due to hypervolemia. This ~10% excess weight gain resolved by days 10–12, after which body mass increased in parallel to control transplants (Fig. 7i). After an initial drop

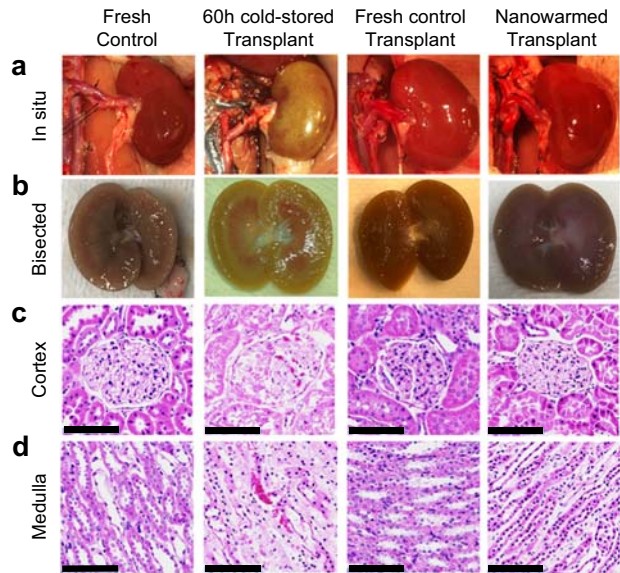

**Fig. 6 | Gross and histologic appearance of nanowarmed and control kidney transplants.** Representative appearance of fresh control kidney before cannulation and recovery (*n* = 5), 60-h cold-stored kidney at the time of intraoperative transplant organ failure (*n* = 1), fresh control kidney transplants at day 30 post transplant (*n* = 5), nanowarmed kidney transplants at day 30 post transplant (*n* = 5). **a** Gross images in situ. **b** Bisected kidneys following explant. **c** Histology of renal cortex (H&E). **d** Histology of renal medulla (H&E). Scale bars are 100 μm. H&E hematoxylin and eosin.

in hemoglobin in both groups due to surgical blood loss, hemoglobin rose steadily in the postop period, suggesting intact renal erythropoietin production and/or potentially hemoconcentration (Fig. 7h).

At the end of the planned posttransplant follow-up (postop day 30), animals were sacrificed for serum and urine analyses and histology. Both serum and urine laboratory parameters demonstrated statistically similar, and essentially normal, kidney function in both groups (Tables 1 and 2). Specifically, creatinine, blood urea nitrogen, and electrolytes were within normal limits for rats[27]. Urinalysis showed the absence of hematuria, pyuria, glycosuria, or proteinuria. The estimated glomerular filtration rate (eGFR) for nanowarmed kidney recipients was 2.2 ± 0.8 mL/min, which was not statistically different from control kidney transplant recipients (2.7 ± 0.7 mL/min, *P* = 0.421) and in the range for normal rats (median: 1.5 mL/min, interquartile range: 1.0–2.2 mL/min[28]). Normal liver function tests and creatinine kinase levels suggested a lack of systemic toxicity, but there was a mild elevation in serum amylase in the nanowarmed organ recipients. Of note, in these transplants there was no correlation of preservation time with peak Cr, time to normalization of Cr to <2.0, or terminal Cr (all *P* > 0.5, Kendall rank correlation).

For comparison, we performed limited testing of a long-term cold stored kidney (60 h in UW solution at 4 °C) transplant. The 60-h time point was chosen based on historical studies demonstrating a clear increase in kidney injury, with a decline in viability and graft function, occurring between 48 and 72 h of cold storage, depending on the preservative solution[29]. The 60-h cold stored kidney reperfused poorly with patchy ecchymosis over the surface and hilar congestion (Fig. 6). Failing to make urine, the rat was euthanized, and the gross appearance of the bisected kidney demonstrated congested medulla and ecchymosis in the cortex.

### Microscopy of nanowarmed and transplanted kidneys
Nanowarmed and fresh control transplant kidneys were recovered at posttransplant day 30 and compared morphologically and histologically to untreated fresh control and 60-h cold stored and transplanted

organs (Fig. 6). The transplanted nanowarmed and transplanted control organs were bisected. Both groups showed grossly intact architecture of the renal cortex, medulla, and pelvis (Fig. 6b). Histologic examination of transplanted nanowarmed kidneys demonstrated some focal tubular necrosis and hyaline change but intact basement membranes and vasculature (Fig. 6c, d). Microthrombi and infarcts were absent. There was a trace amount of yellowish-brown material in some of the thin-walled capillaries of most of the nanowarmed kidneys, best seen at the cortico-medullary junction and medulla but absent in glomeruli, which could represent a small amount of retained IONPs. Fresh control transplants also showed focal tubular necrosis in some areas with some hyaline change but otherwise normal histology. Large blood vessels and collecting ducts were normal in both nanowarmed and fresh control kidneys. The 60-h cold stored kidney transplants demonstrated mesangial hypercellularity and diffuse proximal convoluted tubule necrosis with congestion in the glomerulus and tubules in the medulla. Collecting ducts and larger blood vessels were normal. Further histologic details are in the Supplementary Information.

## Discussion
Successful cryobanking of human organs prior to transplant would revolutionize how organs are recovered, allocated, and ultimately used to cure end-stage organ disease. Organ banking would improve donor/recipient matching, allow for better patient preparation and scheduled procedures, facilitate tolerance induction protocols in recipients, and increase organ utilization—all while supporting graft and patient survival. Here we show the first repeatable approach for successful cryopreservation of intact organs (rat kidneys) for up to 100 days prior to transplantation. Nanowarmed organs restored renal function and solely sustained the lives of nephrectomized transplant recipients for 30 days post transplant. These results show that prolonged organ banking for transplantation may finally be possible. While we demonstrate success in the rat kidney, our approach is translatable to other organs, and the intrinsic scalability of nanowarming suggests clinical translation is feasible.

In this study, we found an initial period of nanowarmed kidney graft dysfunction lasting 2–3 weeks, following which renal function normalized in all nanowarmed kidney transplant recipients. At the 30-day endpoint, renal function of nanowarmed kidney recipients was statistically similar to fresh control transplants, as measured by serum chemistries and urinalyses, and both groups were in the normal range for healthy control rats. Further, recovery of hemoglobin levels suggested intact renal erythropoietin production (and potentially hemoconcentration), and normal calcium levels suggested preserved renal conversion of vitamin D to its active form (1,25-dihydroxyvitamin D). The absence of glycosuria, hematuria, or proteinuria also supported the lack of renal damage. These data demonstrated physiologic renal recovery.

However, it is essential to consider the degree of initial graft dysfunction observed (i.e., peak and width of postoperative creatinine curve) and how this might predict long-term organ function. While we did not follow these recipients beyond 30 days, we can extrapolate potential long-term outcomes from clinical scenarios in human kidney transplantation. From the normothermic perfusion experiments, we found that nanowarmed kidneys produced the same amount of urine as fresh control kidneys (albeit with a slightly reduced creatinine clearance) and performed similarly to the 24-h cold stored control group for all measures. In the transplant experiments, nanowarmed kidneys had higher peak creatinine and took longer to achieve normal levels than fresh control transplants, but both groups performed much better than 60-h cold stored ones. Our nanowarmed kidney recipients had normal creatinine at the end of the 30-day follow-up, which in clinical transplantation predicts good long-term graft function in human kidney recipients[30]. Other predictors of long-term renal

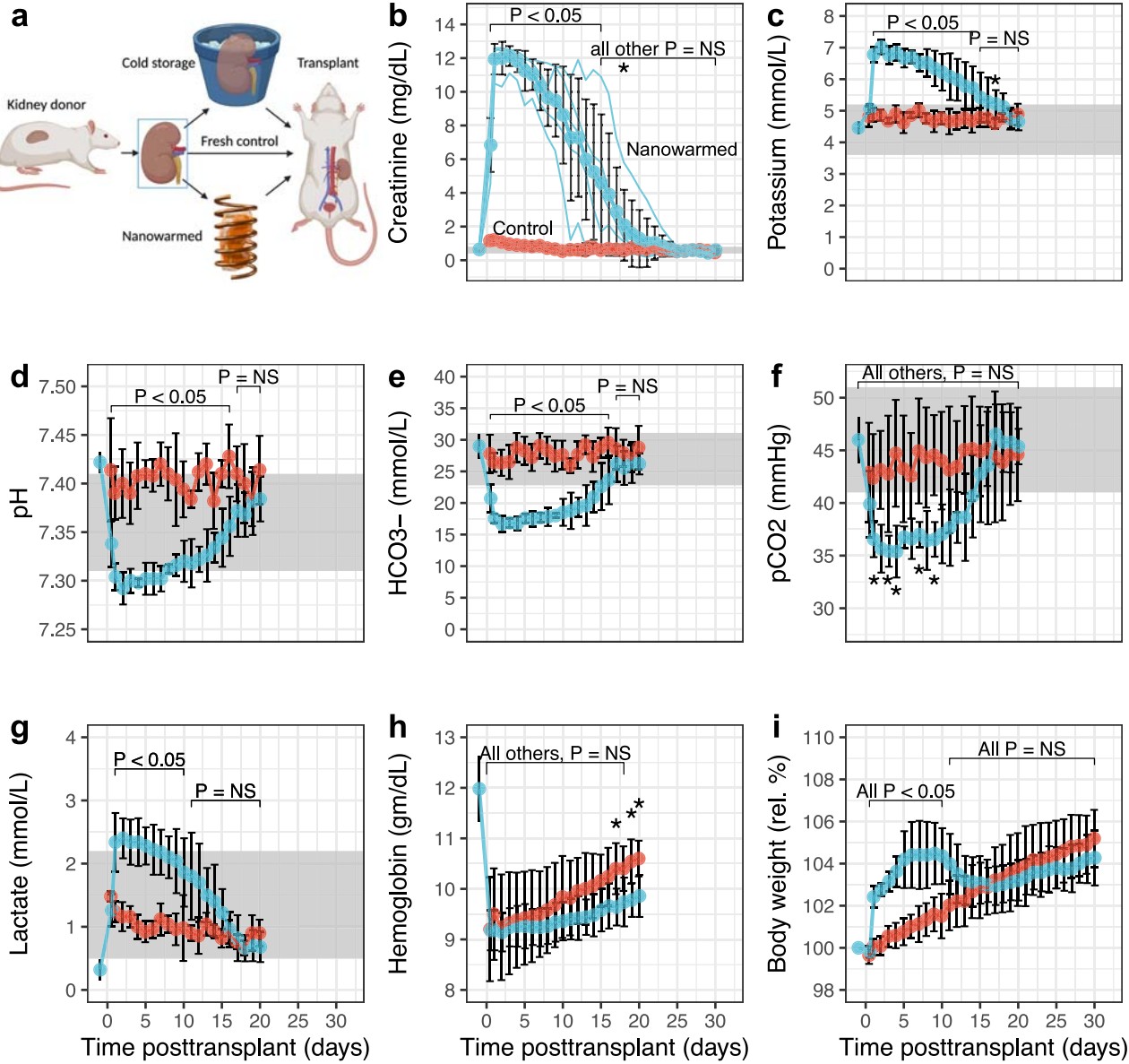

**Fig. 7 | Transplant outcomes of nanowarmed and fresh control kidney transplants. a** Depiction showing the transplant schema. Organs were recovered from donor rats and either immediately transplanted ($n = 5$); vitrified, nanowarmed and transplanted ($n = 5$); or stored for 60 h ($n = 1$) in UW solution prior to transplant. The 60-h cold stored kidney failed intraoperatively, and the recipient was sacrificed. Nanowarmed (blue) and fresh control kidney transplants (red) were followed for 30 days post syngeneic transplant with daily labs and assessments. **b** Serum creatinine with individual recipient curves and aggregate data shown; **c** Venous potassium; **d** Venous pH; **e** Venous bicarbonate ($HCO_3^-$); **f** Venous partial pressure of $CO_2$ (pCO2); **g** Venous lactate; **h** Hemoglobin. **i** Daily weight normalized to day −1 pretransplant. The gray bands are normal ranges. Data are mean ± s.d. Statistical tests were Wilcoxon (**b, d, e, f, g, i**) or $t$-test (**c, h**). *$P < 0.05$, **$P < 0.01$, ***$P < 0.001$; NS not significant. Source data are provided as a Source Data file. Full statistical treatment in Supplementary materials.

function are discussed in the Supplementary Discussion. Altogether, we infer that (1) fresh control transplants in rats are analogous to living donor transplants in humans (best outcomes), (2) 24-h cold stored kidneys are similar to standard criteria deceased donor transplants, and (3) 60-h cold stored kidneys are at the limits of expanded criteria transplants. As such, we can speculate that nanowarmed kidneys will have long-term outcomes similar to those seen in standard deceased donor transplantation.

We note that Fahy and colleagues first demonstrated vitrification of rabbit kidneys in 1984[11] and reported a single successful rabbit kidney transplant[31], but have never reproduced that finding. Their studies were critical for proof-of-concept, but they were limited by their rewarming technique (convective warming on the organ surface), which was unable to reproducibly achieve the needed rates or homogeneity of heating required. Due to limitations in the convective

warming rates, a higher concentration of CPA was required (9.3 M M22) with more significant toxicity than VMP. To overcome that toxicity, they reduced the loading temperature to −22 °C, which slowed CPA delivery due to increased viscosity and decreased diffusivity and membrane permeability. Inadequate tissue CPA concentration, combined with suboptimal rewarming, led to ice formation, particularly in the renal medulla, and failure in almost all transplants[21,31]. Additionally, their single surviving rabbit kidney transplant was vitrified for only 8 min prior to rewarming, whereas we demonstrated stable storage for up to 100 days. Further, their transplant recipient had a nadir creatinine of ~4 mg/dL and a terminal creatinine of ~6 mg/dL, which was much higher than what we observed (nadir 0.4 mg/dL and terminal 0.6 mg/dL, each in the normal range for rats). While their transplant did sustain the life of the recipient, it did so with significantly impaired renal function and the need for repeated blood

**Table 1 | Serum chemistries 30 days post transplant**

| Serum | Control, N = 5[a] | V–N, N = 5[a] | P value[b] |
|---|---|---|---|
| Renal labs | | | |
| BUN (mg/dL) | 19.20 (1.79) | 18.40 (2.41) | 0.7 |
| Creatinine (mg/dL) | 0.44 (0.19) | 0.62 (0.24) | 0.3 |
| Na+ (mmol/L) | 141.00 (3.16) | 141.20 (1.64) | >0.9 |
| K+ (mmol/L) | 4.64 (0.32) | 4.66 (0.50) | >0.9 |
| Cl– (mmol/L) | 102.00 (1.41) | 102.00 (2.35) | 0.8 |
| HCO3– (mmol/L) | 25.06 (1.99) | 25.02 (1.08) | >0.9 |
| Ca2+ (mg/dL) | 9.58 (0.45) | 9.82 (0.36) | 0.2 |
| Mg2+ (mg/dL) | 1.98 (0.11) | 1.76 (0.37) | 0.2 |
| Phos (mg/dL) | 4.56 (0.82) | 4.50 (0.66) | >0.9 |
| Extrarenal labs | | | |
| Total protein (g/dL) | 6.22 (0.63) | 6.76 (0.74) | 0.3 |
| Albumin (g/dL) | 3.66 (0.61) | 4.04 (0.93) | 0.6 |
| Globulin (g/dL) | 2.72 (0.43) | 2.84 (0.23) | 0.5 |
| CK (U/L) | 174 (12) | 147 (17) | **0.032** |
| Total Bilirubin (mg/dL) | 0.18 (0.08) | 0.52 (0.42) | 0.2 |
| ALP (U/L) | 150 (36) | 149 (47) | 0.8 |
| GGT (U/L) | 3.54 (0.67) | 3.10 (0.34) | 0.4 |
| ALT (U/L) | 39 (11) | 47 (17) | 0.8 |
| AST (U/L) | 63 (17) | 53 (31) | 0.7 |
| Amylase (U/L) | 569 (85) | 807 (177) | **0.016** |

Statistically significant P values are in bold.
[a]Mean (SD).
[b]Wilcoxon rank sum test; Wilcoxon rank sum exact test.

**Table 2 | Urinalysis 30 days post transplant**

| Urine | Control, N = 5[a] | V–N, N = 5[a] | P value[b] |
|---|---|---|---|
| Specific gravity | 1.028 (0.013) | 1.022 (0.008) | 0.5 |
| pH | 7.50 (0.61) | 6.88 (0.85) | 0.3 |
| Protein (mg/dL) | 7.78 (0.79) | 8.28 (0.67) | 0.6 |
| Protein | | | |
| Neg | 5 (100%) | 4 (100%) | |
| Glucose | | | |
| Neg | 5 (100%) | 4 (100%) | |
| Urobilinogen | | | |
| Neg | 5 (100%) | 4 (100%) | |
| Ketones | | | >0.9 |
| Neg | 1 (25%) | | |
| Trace | 5 (100%) | 3 (75%) | |
| Occult blood | | | |
| Neg | 5 (100%) | 4 (100%) | |
| RBC (per HPF) | | | |
| 0–5 | 4 (100%) | 3 (100%) | |
| WBC (per HPF) | | | |
| None | 4 (100%) | 3 (100%) | |
| Bacteria | | | |
| Not seen | 4 (100%) | 3 (100%) | |
| Casts | | | |
| Not seen | 4 (100%) | 3 (100%) | |
| Crystals | | | |
| Not seen | 4 (100%) | 3 (100%) | |

[a]Mean (SD); n (%).
[b]Wilcoxon rank sum test; Wilcoxon rank sum exact test; Pearson's Chi-squared test.

transfusion. Further, we can predict that their convective rewarming approach will be insufficient for human-sized organs. In contrast, nanowarming is independent of size.

Other methods for warming cryopreserved organs have been attempted, but with limited success. Direct electromagnetic rewarming of cryopreserved materials, such as dielectric heating at radio and microwave frequencies, uses high-frequency electromagnetic fields to stimulate the oscillation of the polar molecules, thereby converting electromagnetic energy into heat[32]. These approaches have failed to produce adequately rapid and uniform rewarming due to variations in the material dielectric properties, attenuation of the higher frequency fields (e.g., caused by skin depth), and the shape of the sample[33]. For example, microwave rewarming leads to thermal hotspots from limited tissue penetration and standing wave formation[34,35]. Another group attempted nanowarming in rat hearts but did not test for viability or function after rewarming[20]. Another approach called "directional freezing" was tested for liver cryopreservation with partial success, but with only limited biologic outputs (including no survival transplants) and absent follow-on study[36].

Further, a range of other technologies are in use or are being studied to prolong preservation times for kidneys and other organs, including hypothermic machine perfusion[37], normothermic machine perfusion[25], subnormothermic machine perfusion[38], high subzero supercooling[39,40], and partial freezing[41]. While these extend preservation for broader organ sharing, they only increase preservation by hours to a few days and do not achieve prolonged banking for months and potentially years. These approaches, and ours, allow for national organ sharing, more time to perform cross-matching, and converting transplants to daytime events. However, nanowarming uniquely would improve donor/recipient matching by increasing the number of organs available for consideration at one time, whereas other approaches continue the practice of one at a time decision making for every potential organ offer. Further, nanowarming would allow better patient preparation prior to surgery, enable tolerance protocols that

take days to weeks of conditioning or preparation, and increase organ utilization by banking organs even if an appropriate recipient has not yet been identified.

Success in prolonged organ preservation has tremendous potential to transform human organ transplantation but will also require significant changes in the current organ transplantation system. The current system is based on severe time constraints from organ acquisition to transplant, typically measured in hours. This has contributed to a U.S. system characterized by wide variations in organ availability by region and transplant center, inefficient matching of organs and recipients, significant inequities in organ availability, and excessive organ nonuse[42]. Successful banking and relief from time constraints will invite a transformation in the system to address these problems and benefit patients. Additionally, the ethical, legal, and social implications of this technology would be considerable. In particular, quality and safety issues will be forefront in developing a framework for governmental and regulatory oversight.

Our study does have several important limitations. First, it was performed in a rat model with limited sample size (n = 4–8 per group) rather than in human or human-sized (e.g., porcine) kidneys. Second, we only studied organ storage for up to 100 days. Third, we only followed transplant recipients to our 30-day endpoint and did not assess longer-term survival. Fourth, we performed only a limited number of allogenic transplants and have not characterized any changes to the host immune response. Fifth, we have not determined which mechanisms and modes of injury (e.g., apoptosis vs. necrosis, protein denaturation from high concentration CPA solutions) led to the observed initial transplant dysfunction in the nanowarmed kidneys. Sixth, we have not fully characterized how nanowarmed organs perform compared to other categories of transplants, such as 36–48 h of cold storage. And finally, while the organs eventually recovered fully, they experienced a period of 2–3 weeks of initial graft

dysfunction, which might have been treated with dialysis in some clinical settings.

Future directions to address those limitations and other questions include scaling up to human-sized kidneys, conversion to clinical-grade reagents and processes, thorough investigation of the modes and mechanisms of injury (e.g., apoptosis and mitochondrial content and function) to develop injury mitigation strategies, characterizing preservation limits and durability, and defining the host immune response.

## Methods

Our study complies with all relevant ethical considerations. The Institutional IACUC committee from the University of Minnesota (protocol #2204-39970 A) approved all animal studies.

### Animals

450–525 gm, 16–32-week-old, male Lewis (Strain #004) and Sprague Dawley rats (Strain #400) were purchased from Charles River Laboratories. Lewis rats were used as donors and recipients for syngeneic transplants and Sprague Dawley for allogeneic transplants and in vitro studies. Male sex was selected for their larger size and to avoid Y chromosome encoded-antigen immunoreactivity. Rats were housed in a conventional housing facility with a 12-h on/12-h off light cycle, 68–74 °C ambient temperature, 30–50% humidity, and free access to food (Envigo Lab Diet #2918) and water.

### CPA solutions and nanoparticles

VMP (16.8 wt% ethylene glycol, 22.3% DMSO, 12.9% formamide, 1% X-1000, 1% Z-1000) and VS55 (16.8 wt% propylene glycol, 24.2% DMSO, 14.0% formamide, 0.24% HEPES) were prepared in carrier solutions previously used[18,21], except that the carrier for VMP experiments (LM5-XZ) was modified from the original LM5[43] by adding synthetic ice blockers 1% X-1000 (w/v) and 1% Z-1000 (w/v) (21st Century Medicine, Fontana, CA). For these experiments, we used silica-coated iron oxide nanoparticles (IONPs) that were synthesized from EMG308 iron oxide core nanoparticles (Ferrotec, Santa Clara, California) that were coated with a silica shell and surface modified with polyethylene glycol and a small hydrophobic ligand, trimethylsilane[22], suspended in CPA (VMP), and filtered before use. To avoid misinterpretation of experiments where IONPs may have aggregated after filtering, we set prospective exclusion criteria wherein the perfusion flow rate during unloading had to be >45% of the observed flow rate during loading.

### Kidney cannulation and recovery

Rats were anesthetized with isoflurane, and a cruciate laparotomy was performed. The bowel and mesentery were retracted. The abdominal aorta, inferior vena cava, and left renal artery and vein were mobilized for cannulation. The kidney was dissected free of Gerota's fascia. A 20G bulb tip catheter (FTP-20-30, Instech Laboratories, Plymouth Meeting, PA) was used to cannulate the abdominal aorta below the renal arteries. A second cannula (Male Luer to Hose Barb Adapter, 45518-46, Cole Parmer, Vernon Hills, IL) was used to cannulate the inferior vena cava for venous drainage. A third cannula (PE-10-100 polyethylene tubing; 0.011″ ID × 0.025″ OD; SAI Infusion Technologies) was used to cannulate the ureter. The suprarenal aorta was cross clamped, and 15 mL of University of Wisconsin (UW) solution containing 500 IU of heparin was perfused at 0–4 °C. The suprarenal aorta and vena cava were ligated and divided above the left renal vessels. The kidney was explanted and stored in UW at 4 °C until use (30 min to 2 h).

### Perfusion CPA loading

Kidneys were connected to a custom-built multithermic perfusion system for pressure- or flow-regulated perfusion[17,18]. When VS55 was used as the CPA, ramp loading was performed as previously reported[44].

When VMP was used as the CPA, we first used the loading protocol developed for that CPA[21], but changed to a modified protocol as follows. The kidney was first flushed for 20 min with a modified carrier solution (diluent in preservation solution in which CPA cocktails are prepared). In this case, the carrier was LM5-XZ, based on the original VMP carrier (LM5)[21], but containing synthetic ice-blockers X-1000 and Z-1000[43]. This solution helped improve vascular perfusion due to its oncotic properties. We then introduced VMP by ramping the concentration from 0 to 5 M by 50 mM/min, during which the flow rate decreased due to rising viscosity. At 5 M, we held the concentration steady for 10 min to allow for osmotic equilibration and then stepped up to full-strength VMP (8.4 M) to minimize exposure times at the highest CPA concentrations. Simultaneously, we increased the perfusion pressure from 40 to 60 mmHg to maintain adequate flow rates during the 25 min full-strength VMP perfusion. The perfusion loading of VMP took 155 min in total.

### Loading of nanoparticles

Following the last step of VMP loading, 10 mg Fe/mL IONPs in VMP was perfused via syringe pump at 4 °C. Perfusion duration was 4–5 min at a constant flow rate of 0.5 mL/min, and the loading pressure was monitored[18]. After perfusion, the kidney was disconnected and placed in a 2 × 3-inch polyethylene bag (McMaster-Carr, Elmhurst, IL) prefilled with ~20 mL of VMP with 4 mg Fe/mL IONPs at 4 °C. A fiberoptic temperature probe (Qualitrol, Fairport, NY) was placed in the solution adjacent to the kidney to record the temperature, and a data logger (Qualitrol T/Guard, Fairport, NY) was used to record the thermal history at 1 s intervals.

### Mass transport model

CPA and water transport inside the kidney were simulated by Krogh cylinder model[45]. In the Krogh model, the kidney is represented by many parallel identical cylindrical/hexagonal prism units, each consisting of a central capillary surrounded by tissue. The mass transfer in the whole kidney is analyzed by considering the processes in such a unit to be typical and representative of the processes in the whole organ. The transport from the capillary vasculature into the extravascular space is assumed to occur across a membrane (composite basement and cell membrane) that can be described by coupled solute/solvent flow using irreversible thermodynamics as described by the Kedem-Katchalsky (KK) equations. Three important parameters used in the KK equations are $L_p$ (hydraulic conductivity, [$m^3/(N\,s)$]), $\omega$ (CPA permeability, [$mol/(N\,s)$]), and $\sigma$ (reflection coefficient). The values of those parameters were $L_p = 1.5 \times 10^{-14}$ $m^3/(N\,s)$, $\omega = 7.0 \times 10^{-13}$ $mol/(N\,s)$ and $\sigma = 0.1$, which were determined in preliminary experiments. The combined Krogh cylinder and KK modeling were applied to the previous VMP loading protocol[21] and the modified protocol developed in this study.

### Vitrification of kidneys

Vitrification was achieved in a controlled rate freezer (Kryo 560-16, Planer Ltd, Middlesex, UK). The controlled rate freezer was pre-programmed to start at a chamber temperature ($T_{chamber}$) = 0 °C and cool down to $T_{chamber} = -122$ °C at a ramp rate of $-40$ °C/min. A 25 min annealing step was introduced when the chamber reached $-122$ °C (just above the glass transition temperature ($T_g$)) to allow the organ to thermally equilibrate and reduce the introduction of thermal stress before the glass transition. After the annealing step, a slower ramp rate of $-5$ °C/min was used to cool from $-122$ °C down to $-150$ °C, reducing the build-up of thermal stress as the organ entered and cooled in the glassy phase. At $T_{chamber} = -150$ °C, a 10-min temperature-hold step was used to equilibrate the organ to the storage temperature ($-150$ °C) before rapid transfer to a $-150$ °C freezer (PHC Corporation of North America, Wood Dale, IL) for storage until rewarming.

## Nanowarming of kidneys

Nanowarming was conducted using a 15 kW radiofrequency (RF) coil (AMF Life Systems, Auburn Hills, Michigan) at 94% power (provides an RF field at 63 kA/m and 180 kHz with field variation ≤ ±5% across the ~80 mL coil bore)[14]. The bag containing the vitrified kidney was transferred from the −150 °C storage freezer to a Styrofoam container chamber containing liquid nitrogen and vapor (which maintained the temperature near −150 °C for > 30 min) to move the kidney to the RF device. The sample was then transferred from the Styrofoam container to the center of the coil, and the alternating magnetic field was switched ON to initiate nanowarming. A data logger was used to record the thermal heating history. The field was switched OFF when the temperature reached −25 °C (above the melting temperature of VMP). The bag containing the kidney was removed from the coil, placed on ice, and CPA unloading began within 3 min.

## Perfusion unloading of kidneys

The kidney was reconnected to the perfusion system and unloading started by perfusing 4.2 M VMP with 300 mM mannitol for 15 min. Mannitol was added to reduce cell swelling during exposure to the relatively hypotonic unloading solutions. Then the concentration of CPA was ramped down from 4.2 M VMP with 300 mM mannitol to 0 M (LM5-XZ) over 120 min. The VMP ramping rate was −35 mM/min, and the mannitol ramping rate was −2.5 mM/min. The kidney was then flushed with LM5-XZ for 30 min. The perfusion unloading took 165 min in total, and pressure was kept at 40 mmHg and temperature at 0-4 °C for the entire process. After unloading, the kidney was flushed with cold UW hypothermic preservation solution, disconnected from the perfusion circuit, and placed back in UW solution on ice.

## Micro-CT imaging and histology

Micro-CT was performed at a resolution of 0.061 mm. Briefly, the kidneys were scanned on a micro-CT imaging system (NIKON XT H 225, Nikon Metrology, MI) with an accelerating voltage of 65 kV and current of 95 μA[46]. The vitrified kidney in a cryobag was held in LN2 vapor (−150 °C) in a Styrofoam container during imaging. Separate tubes of water and air at room temperature were attached to the top of the container to serve as calibration references for determining radiodensity in Hounsfield units (HU). The images were reconstructed to reduce beam hardening artifacts and improve image quality (3D CT pro, Nikon Metrology, MI). The images were converted to unsigned 16-bit float images, post-processed (VGstudio Max 3.2, Volume Graphics, NC), and exported as DICOM images for a final analysis using MATLAB (MathWorks).

Histology with hematoxylin and eosin (H&E) or Periodic acid−Schiff (PAS) was also performed as routine[18]. The kidney slices were digitized for histopathological analysis, and histologic interpretation was performed in a blinded fashion by a clinical pathologist.

## Differential scanning calorimetry

A differential scanning calorimeter (DSC, Model Q1000, TA Instruments, New Castle, DE) was used to record thermograms (heat flow vs. temperature) of crystallization events in samples during cooling and rewarming. All calculations for DSC measurements are performed in the thermal analysis software (Universal Analysis 2000, TA Instruments). A parameter (i.e., ice fraction (%)) was primarily calculated at a given cooling and/or warming rate, as described previously[47]. CCR and CWR were calculated from the measured heat flow using ice fractions following previously published methods[48,49]. $T_m$ (peak) and $T_g$ (inflection point) of heating were calculated using the heating cycle of DSC heat flow curves thermogram[50].

## Normothermic machine perfusion

Normothermic machine perfusion of kidneys was performed as we have reported for livers[19] with brief modifications. The arterial perfusate was a modified Krebs−Henseleit Bicarbonate Buffer (KHB, 120 mM NaCl, 4.7 mM KCl, 1.2 mM $KH_2PO_4$, 1.5 mM $CaCl_2$, 1.2 mM $MgSO_4$, 25 mM $NaHCO_3$, 0.1 mM EDTA) that was supplemented with 5.54 mM glucose, 5 g/L BSA, 1 g/L amino acids, and 0.5 g/L creatinine. Pressure was measured with a sensor at the arterial perfusion catheter (PREPS-N-000, PendoTECH, Princeton, NJ), and the flow rate was adjusted to maintain 90-110 mmHg. Oxygenation was achieved with carboxygen (95% $O_2$ and 5% $CO_2$), and the temperature was maintained at 37 °C with an inline heat exchanger. Samples from the vein and ureter were taken at 10-min intervals and analyzed using an ABL90 FLEX Plus point of care system (Radiometer, Brea, CA) and enzymatic creatinine assay (Rat Creatinine Kit #80340, Crystal Chem, IL, USA). Oxygen consumption rate was calculated as described[51], creatinine clearance as described[52], and glucose consumption rate was calculated as (arterial glucose concentration − venous glucose concentration) × arterial flow/kidney weight.

## Kidney transplantation

Inbred male Lewis rats (450–525 g) were used as donors and recipients for syngeneic transplants, and outbred Sprague Dawley rats (450–525 g) were used as donors and recipients in a limited number of outbred rat transplants. Baseline body mass and laboratories were obtained on pre-operative day −1. Recipient rats were anesthetized with isoflurane, and a laparotomy was performed. The bowel and mesentery were mobilized and retracted. The abdominal aorta, inferior vena cava, and left renal artery and vein were dissected free of adherent tissue. The left native kidney was mobilized free of Gerota's fascia and skeletonized down to the hilar vessels. The left native ureter was transected close to the hilum for maximum length. The left renal artery was ligated and divided. A microvascular clamp was placed on the renal vein, and the vein was divided to explant the native kidney.

Two arterial microvascular clamps were placed proximal and distal to the anastomotic site on the recipient's infrarenal aorta. An arteriotomy was performed and immediately flushed with heparinized saline. The donor kidney was flushed with cold normal saline, wrapped in cold gauze, lowered into the field, and periodically dripped with cold saline to maintain hypothermia until reperfusion. An end-to-side (donor end aorta to recipient side aorta) anastomosis was performed using running 10−0 Prolene suture. The venous anastomosis was then performed either using the cuff technique[53] in an end-to-end fashion with the native left renal vein or by an end-to-side sutured anastomosis between the donor renal vein and the inferior vena cava. Once the arterial and venous anastomoses were complete, the venous and arterial clamps were released to reperfuse the kidney. Warm normal saline was poured onto the reperfused kidney. After visualization of urine in the catheter, the right native renal artery, vein, and ureter were ligated and divided, and the native right kidney was explanted. An end-to-end anastomosis of the donor and recipient ureter was performed over a PE-10 stent. The abdomen was closed in layers using 4-0 PDS in the abdominal wall and 5-0 PDS in the skin. The rat was incubated on a heating pad until recovery from anesthesia while providing supplemental oxygen.

Venous blood gas was measured daily from day −1 to day +20 (ABL90 FLEX Plus). Serum creatinine was measured daily from day −1 to day +30 (Rat Creatinine Kit), as was body weight. The rats were euthanized on postoperative day 30. The kidneys were recovered for histology, and serum and urine were collected for analysis by a clinical Vet Med laboratory. The estimated glomerular filtration rate (eGFR) was calculated as described[28].

## Statistical analysis

Statistical analysis was performed in R version 4.2.1 (R Foundation for Statistical Computing, Vienna, Austria). The number of biological replicates is indicated in each figure legend and the accompanying

statistical treatment summary in the Supplementary materials. All measurements represent distinct biological replicates taken from individual kidneys or rats, except for time series data, where kidneys/rats were resampled at each time point. For comparison of continuous variables, normality was established using the Shapiro–Wilk test or qq plots and distribution histograms. Homogeneity of variance was assessed using Levene's test. For normal (or near-normal) group comparisons, ANOVA testing with pairwise post hoc *t*-test for single comparisons, Tukey HSD test for multiple comparisons with equal variance, or the Games–Howell test if unequal variance were used to determine statistical differences. Non-normal variables were tested using the non-parametric Kruskal–Wallis test for overall significance and the pairwise Wilcox (Mann–Whitney *U*) test for individual group comparison. Categorical variable comparison was performed via Pearson's Chi-squared test. *P* values were adjusted for multiple comparisons. Complete statistical treatment for each figure is presented in the supplementary files (Supplementary Data 1). Statistical testing was two-sided, and a *P* value of <0.05 was considered significant.

### Reporting summary

Further information on research design is available in the Nature Portfolio Reporting Summary linked to this article.

## Data availability

All data generated or analyzed during this study are included in this published article (and its supplementary information files). Source data are provided with this paper.

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

## Acknowledgements

Z.H. and J.S.R contributed equally as co-first authors and are listed in alphabetical order. J.C.B. and E.B.F. contributed equally as co-senior and corresponding authors. We acknowledge Dr. Usha Kini (St. John's National Academy of Health Sciences, India) for interpreting histologic samples; Diane Tobolt, Mikaela Hintz, and Saurin Kantesaria for technical assistance; Andy Grams for technical illustration; Dr. Michael Lotti for editorial assistance; and Dr. Greg Fahy for scientific discussion on CPA formulation. Some figures were created with BioRender.com (Figs. 1, 4, and 7). This work was supported by NIH grants DK117425 (E.B.F, J.C.B.), HL135046 (E.B.F, J.C.B.), DK131209 (E.B.F, J.C.B.), and DK132211 (E.B.F, J.C.B.), NSF grant EEC-1941543 (E.B.F, J.C.B., T.L.P., S.M.W.), and a gift from the Biostasis Research Institute funded in part through contributions from LifeGift, Nevada Donor Network, Lifesource, Donor Network West, and Lifebanc.

## Author contributions

Z.H. performed the CPA loading and unloading, vitrification and nanowarming, micro-CT experiments, and mass transport modeling. He also prepared figures and edited the paper. J.S.R. performed the surgical procedures, normothermic machine perfusion, kidney transplants, and histology. He also prepared figures and edited the paper. B.N. assisted with the development of the surgical model and edited the paper. J.P.A. refined nanoparticle synthesis, production, CPA stability, and delivery and edited the paper. L.G. performed some experiments and edited the paper. M.L.E. managed the project, performed some experiments, and edited the paper. S.M.W. and T.L.P. edited the paper and added content. E.B.F. and J.C.B. conceived and supervised the project, analyzed the data, prepared figures, and wrote and edited the paper.

## Competing interests

The authors declare the following intellectual property related to this work: "Cryopreservative compositions and methods" Pending U.S. Patent Applications 14/775,998 and 17/579,369 (M.L.E and J.C.B.). All other authors declare no competing interest.
