## [Peer Review File · Nature Communications]

Vitrification and nanowarming enable long-term organ cryopreservation and life-sustaining kidney transplantation in a rat modelREVIEWER COMMENTS

Reviewer #1 (Remarks to the Author):

The paper - Enabling organ banking: successful kidney cryopreservation followed by life sustaining transplant is a very important contribution to the field of organ and tissue banking. The ability to transplant organs (here rat kidneys) with long term survival and in a consistent way has been attempted for more than 40 years, but until now it has not been achieved. The manuscript is clearly written and the complex methods have been well described. proper attention has been paid to ethical factors involved in the work and to the statistical evaluations used. The text makes a comprehensive description for both the cryopreservation technologies and the organ transplantation. This important report is suitable for publication should it meet with editorial final approval. No changes are needed to the text.

The only very trivial comments which could be made to help readers is for more comprehensive descriptions of the pathology images which are shown for the transplanted kidneys. Many readers will not be familiar with kidney morphology, and greater in depth explanations e.g. in figure legends in supplementary files could enhance understanding.

Reviewer #2 (Remarks to the Author):

Han et al report development of an approach to cryopreservation of organs for transplantation that will remove time constraints in organ distribution, improve equity in access and increase utilization while reshaping the current emergency nature of the transplant paradigm. The authors report convincing groundbreaking work demonstrating that with optimized cryoprotective agents (CPAs) and iron nanoparticles in the vasculature with oscillating magnetic field rewarming can achieve long term organ storage by vitrification with full recovery of renal function post transplantation. Importantly, they also suggest that the protocol developed should be scalable to human sized organs. This is fascinating and powerful work with potential clinical applicability.

1. It is clear that the vitrification of rat kidneys and their rewarming/recovery incites some damage to the organ though recovery is seen when survival is out past 3 weeks.

Understanding the nature of this recovery is essential to understand the likely performance on a clinical scale and in more complicated transplant situation with toxic immunosuppression etc. The authors might give this area further attention.

2. I find figure 3, a key data slide for the experiment to be unsatisfying having only 3 cases per group. At least for the key groups additional n would be more compelling.

3. Was the level of tissue ATP measured sequentially during the process of vitrification and storage over time, and during or immediately after organ rewarming. ATP has been found to be relevant in the cold and warm perfusion of other organs, especially those from donors suffering an ischemic insult like those recovered after circulatory death. It would be of interest to see the stability of ATP under these preservation conditions.

4. It is notable and encouraging that the V-N transplants eventually recover normal or near normal function. The slow recovery of function post-transplant however may be problematic given high potassium levels, low pH etc. If V-N was applied widely in clinical practice, would there also likely be a high rate of ATN in kidneys that would have otherwise functioned promptly? Also, it appears that the hemoglobin and body weight never fully recover during the study period. The authors should address these points with longer term studies to understand whether permanent damage has ensued. Furthermore, it may be expected that the parameters assessed in Fig. 5 will likely incur additional compromise in an allogeneic transplant under immunosuppression.

5. There should be a more complete assessment of whether the function of V-N kidneys is altered based on the duration of vitrified storage.

6. Use of normothermic perfusion has the potential for assessing the suitability of organs of

unclear quality. Do the authors envision assessing quality and function pre V-N to avoid preserving organs that will later prove unsuitable?

7. Also, the authors could do a better job distinguishing the benefit of this approach versus that of normothermic perfusion which current data suggest can maintain a stable organ function for a few days. It would seem that a 1-2-day storage duration should be sufficient for most cases to allow elective transplants, patient matching if needed, etc. Can the authors be more specific about what cases would benefit from 3 months storage? Also, the statement that this will usher in a new paradigm "in which kidney and other organs wait for patient rather than patients waiting for organs" is poetic but seems hyperbolic since the technology does not necessarily make more organs available and that there are 100,000 patients waiting. To be fair, there are a significant number of organs lost to logistical issues, but this is a small fraction of those transplanted.

Response to Reviewers

We thank the reviewers for their careful review of our manuscript “**Enabling organ banking: successful kidney cryopreservation followed by life-sustaining transplant.**” We were pleased to see the positive reviews, such as “fascinating and powerful work with potential clinical applicability.” Below we address each of the issues raised during the review. We have revised the manuscript and included excerpts from that revision here and **highlighted** them in the revision.

REVIEWER COMMENTS (Original reviewer comments in *italics*, our responses follow in **blue**)

Reviewer #1 (Remarks to the Author)

The paper - Enabling organ banking: successful kidney cryopreservation followed by life sustaining transplant is a very important contribution to the field of organ and tissue banking. The ability to transplant organs (here rat kidneys) with long term survival and in a consistent way has been attempted for more than 40 years, but until now it has not been achieved.

The manuscript is clearly written and the complex methods have been well described. proper attention has been paid to ethical factors involved in the work and to the statistical evaluations used. The text makes a comprehensive description for both the cryopreservation technologies and the organ transplantation. This important report is suitable for publication should it meet with editorial final approval. No changes are needed to the text.

Response: We thank the reviewer for the supportive comment.

The only very trivial comments which could be made to help readers is for more comprehensive descriptions of the pathology images which are shown for the transplanted kidneys. Many readers will not be familiar with kidney morphology, and greater in depth explanations e.g. in figure legends in supplementary files could enhance understanding.

Response: We have included a complete description of the histologic findings in the revised supplementary results. Please see the section “*Microscopy of control, CPA perfused, and nowarmed kidneys*” in the revised supplementary information and added new Supplementary Data Fig. 4 with more detailed histologic images.

Reviewer #2 (Remarks to the Author)

Han et al report development of an approach to cryopreservation of organs for transplantation that will remove time constraints in organ distribution, improve equity in access and increase utilization while reshaping the current emergency nature of the transplant paradigm. The authors report convincing groundbreaking work demonstrating that with optimized cryoprotective agents (CPAs) and iron nanoparticles in the vasculature with oscillating magnetic field rewarming can achieve long term organ storage by vitrification with full recovery of renal function post transplantation. Importantly, they also suggest that the protocol developed should be scalable to human sized organs. This is fascinating and powerful work with potential clinical applicability.

Response: We thank the reviewer for the supportive comments.

1. It is clear that the vitrification of rat kidneys and their rewarming/recovery incites some damage to the organ though recovery is seen when survival is out past 3 weeks. Understanding the nature of this recovery is essential to understand the likely performance on a clinical scale and in more

complicated transplant situation with toxic immunosuppression etc. The authors might give this area further attention.

Response: We agree that understanding the mechanisms of graft injury and recovery will be necessary for future development and translation. We are working on a follow-up study to understand them further. Future studies will investigate pathways of cell stress (transcriptomics), biochemical injury (mitochondrial dysfunction, membrane injury), and cell death (apoptosis, necrosis). Those studies may lead to improvements in the methodology, such as CPA toxicity minimization or identifying targets for damage mitigation (e.g., apoptosis inhibitors and post-rewarming resuscitation to restore ATP levels). However, those studies are outside the scope of this work, which reports the important breakthrough on its own, as the reviewer already notes.

The reviewer also correctly notes that we have not studied how immunosuppression could affect outcomes following transplant of nanowarmed organs. In clinical transplant, we have evidence that mTOR inhibitors (i.e., sirolimus) may hamper organ recovery after acute injury, whereas calcineurin inhibitors (CNIs) are better tolerated. This is part of the reason that CNIs are used acutely after transplant while mTOR inhibitors have black box warnings against early use. This is certainly an area we plan to investigate more as we get closer to clinical translation.

2. I find figure 3, a key data slide for the experiment to be unsatisfying having only 3 cases per group. At least for the key groups additional n would be more compelling.

Response: We have increased the sample size and revised Fig. 3 accordingly. We agree with the reviewer that the original sample size for the normothermic perfusion experiments may have limited statistical power to discriminate subtle differences between groups. This would have been particularly evident if data were non-normally distributed (which was not generally the case) and required nonparametric testing. To address the concern raised by the reviewer, we performed a power analysis from the original dataset using urine output as a test variable. We included the following assumptions: $\alpha = 0.05$, power ($1 - \beta$) = 0.8, and the sampling ratio is 1:1. We found a minimum sample size of $n = 4$ in each group, and that sample size gave an actual power of 0.91.

With that background, we performed additional experiments to increase the sample size to $n \geq 4$ for each group in the normothermic perfusion experiments (see revised Fig. 3). The curves looked nearly identical to those in the original version, and the additional replicates did not change the data interpretation meaningfully. With the increased sample size, slight changes in the P values made some differences statistically significant that were previously not so. However, the general observations and trends remained consistent (fresh > cold storage ~ VMP only ~ nanowarmed >> VS55 only) for the normothermic perfusion outcome measures (i.e., oxygen consumption rate, urine production). We'd also like to highlight that the intent of this report was to show first of its kind, proof-of-concept for the approach. As described in the response to (1.) above, we plan a more in-depth follow-up study to evaluate the mechanisms of graft injury. This will include a more detailed comparison between groups as we refine the protocol further.

3. Was the level of tissue ATP measured sequentially during the process of vitrification and storage over time, and during or immediately after organ rewarming. ATP has been found to be relevant in the cold and warm perfusion of other organs, especially those from donors suffering an ischemic insult like those recovered after circulatory death. It would be of interest to see the stability of ATP under these preservation conditions.

Response: We did not measure ATP levels in these experiments. Unlike in large animal models, ATP sampling in the rat kidney is a destructive process, and it is not feasible to make serial determinations from individual organs. ATP measurement, as well as other measures of cellular respiration such as mitochondrial membrane potential, cellular respiration, and mitochondrial

ultrastructure will be included in a more in-depth follow-up study (see our recent manuscript for the detailed approach taken to assess pancreatic islets following cryopreservation (Zhan, *Nat Med*, 2022)). Likely, ATP depletion during the VN process is less critical than the ability to regenerate ATP after transplant.

4. It is notable and encouraging that the V-N transplants eventually recover normal or near normal function. The slow recovery of function post-transplant however may be problematic given high potassium levels, low pH etc. If V-N was applied widely in clinical practice, would there also likely be a high rate of ATN in kidneys that would have otherwise functioned promptly? Also, it appears that the hemoglobin and body weight never fully recover during the study period. The authors should address these points with longer term studies to understand whether permanent damage has ensued. Furthermore, it may be expected that the parameters assessed in Fig. 5 will likely incur additional compromise in an allogeneic transplant under immunosuppression.

Response: We thank the reviewer for bringing up these important points of consideration. We will address each point/question below:

- a. *If V-N was applied widely in clinical practice, would there also likely be a high rate of ATN in kidneys that would have otherwise functioned promptly?*

Response: Clinically, it is likely that we would have a higher rate of ATN in nanowarmed kidneys compared with those with short static cold preservation times. In rats, the 2-3 weeks of dysfunction were tolerated without treatment (i.e., diuretics or dialysis). And, as the reviewer notes, renal function normalized after that. In clinical transplants, some patients might require short-term dialysis (delayed graft function). Hypervolemia and hyperkalemia would likely have been lessened with furosemide or other medications.

However, in our rat experiments, the function of nanowarmed kidneys was similar to 24-hour static cold storage when assessed by diagnostic normothermic perfusion. It will be interesting to more precisely determine how these organs compare to other conditions in their short- and long-term outcomes. It is unlikely that nanowarming will improve organs compared to their initial state, but what clinical scenario do they most resemble? 24 hours cold storage? 36 hours of cold storage? Those details remain to be determined and will be part of future studies as we refine the protocol.

- b. *it appears that the hemoglobin and body weight never fully recover during the study period*

Response: We stopped daily blood measurement at day 20 as the metabolic function had normalized in all rats. We then tested serum and urine laboratories at the end of follow-up (day 30). We agree that the hemoglobin had yet to reach pretransplant values by day 20 when we stopped monitoring it. Still, hemoglobin levels maintained an upward trend even after the resolution of hypervolemia. Further, there was no statistical difference between hemoglobin values in control and nanowarmed organ transplants. This suggests but does not formally prove that erythropoiesis was intact which will be more carefully studied in our follow-up study.

Nanowarmed kidney recipients initially exhibited hypervolemia with a 2-3% excess body weight gain compared to control recipients over the first 10 days. After 10 days, the difference in relative weight between nanowarmed and control recipients had resolved, suggesting the resolution of hypervolemia. Both nanowarmed and control kidney transplant recipients gained approximately 4-5% in body weight over the 30-day follow-up, which would be typical growth for rats at that age.

- c. *The authors should address these points with longer-term studies to understand whether permanent damage has ensued.*

Response: Longer-term outcome studies are planned. However, these would likely require 6-12 months of additional follow-up to be informative, which is beyond the allowed timeframe for revision of the paper and beyond the scope of this manuscript. As discussed above, work is ongoing to study mechanisms of injury and potential improvements to the protocol to reduce acute injury (i.e., peak and width of Cr elevation). Nonetheless, several reports in clinical kidney transplants show that good function at 1-3 months post-transplant is a predictor for good long-term graft function. We may infer or extrapolate from those clinical studies that the normalization of renal function seen 2-3 weeks after the transplant of nanowarmed kidneys would predict long-term graft function from these kidneys, but it remains to be thoroughly studied.

d. it may be expected that the parameters assessed in Fig. 5 will likely incur additional compromise in an allogeneic transplant under immunosuppression

Response: This is an excellent point that we plan to address in future studies. Briefly, we do not yet know whether alloimmunity and immunosuppression would negatively impact graft function. Since organ cryopreservation is a new technology, there are no real data on how it intersects with the immune response. Hypothetically, one might expect the acute injury to release antigens and augment the priming of an alloimmune response. However, a single report of islet cryopreservation from the 1980's (Couombe, Diabetes 1987) showed that islet cryopreservation *prolonged* allograft survival. We simply do not know if cryopreservation of whole organs will cause injury that is additive, synergistic, or protective of damage associated with the immune response.

5. There should be a more complete assessment of whether the function of V-N kidneys is altered based on the duration of vitrified storage.

Response: We agree that future studies will be needed to assess preservation limits with this technology, but we found no change in organ function over the preservation durations studied (1-100 days). We provide the following evidence that organs are stable under these cryogenic conditions over timescales of months, if not years:

- Transplants of organs preserved for up to 100 days behaved similarly to those preserved for shorter times (see Table below).
- Our results in other systems, such as pancreatic islets (Zhan, *Nat Med*, 2022) and *Drosophila* (Zhan, *Nat Comm*, 2021), show no decrement in function after 6-9 months of cryopreservation.
- Historical literature suggests that vitrified organs can be theoretically stored indefinitely (Fahy, *Cryobiology*, 1984).
- With that background, we re-analyzed the dataset. We found no correlation between preservation time and post-transplant outcomes (peak Cr, time to normalization of Cr < 2.0 mg/dL, and Cr at day 30 posttransplant). These results are included in the revised manuscript (see Table below).

We added the following to the text: "Of note, in these transplants there was no correlation of preservation time with peak Cr, time to normalization of Cr to < 2.0, or terminal Cr (all P > 0.5, Kendall rank correlation)." Below is a summary table for the correlation analysis:

var1	var2	cor	statistic	p	method
preservation_time_hr	pk_cr	-0.25	-0.5702659	0.568	Kendall
preservation_time_hr	time_to_cr_lt_2	0.00	0.0000000	1.000	Kendall
preservation_time_hr	cr_day_30	-0.11	-0.2526456	0.801	Kendall

6. *Use of normothermic perfusion has the potential for assessing the suitability of organs of unclear quality. Do the authors envision assessing quality and function pre V-N to avoid preserving organs that will later prove unsuitable?*

Response: Pre- and post-preservation assessment and conditioning could be helpful in a clinical setting, but we did not include them in the current study. Here we use acellular NMP mainly as a diagnostic for organ function post nanowarming. As the reviewer suggests, NMP could help assess an individual organ's function before transplant and help resuscitate organs to reduce initial dysfunction. Additionally, machine perfusion may provide a “final mile” logistic solution for organ distribution. Organs could be rewarmed at a regional center and then transported to individual transplant centers while being perfused. Thus, as the reviewer alludes, nanowarming and normothermic perfusion could be used synergistically.

7. *Also, the authors could do a better job distinguishing the benefit of this approach versus that of normothermic perfusion which current data suggest can maintain a stable organ function for a few days. It would seem that a 1-2-day storage duration should be sufficient for most cases to allow elective transplants, patient matching if needed, etc. Can the authors be more specific about what cases would benefit from 3 months storage? Also, the statement that this will usher in a new paradigm “in which kidney and other organs wait for patient rather than patients waiting for organs” is poetic but seems hyperbolic since the technology does not necessarily make more organs available and that there are 100,000 patients waiting. To be fair, there are a significant number of organs lost to logistical issues, but this is a small fraction of those transplanted.*

Response: We apologize to the reviewers for not clearly describing some of the relative benefits of long-term cryopreservation over other organ preservation approaches that extend preservation time to 1-3 days (i.e., normothermic machine perfusion or high sub-zero preservation). There are benefits common to each of these approaches (daytime transplants and national organ sharing), but cryopreservation for months or years offers several additional advantages:

- Improved donor/recipient matching. While extending preservation by 1-2 days does allow additional time for testing donor/recipient compatibility (cross-matching and donor specific antibodies) and transplant of organs across the country, it still maintains the one-at-a-time offer/acceptance decision practice. According to current practice, the kidneys will be offered to recipients on the waiting list according to the current UNOS allocation priority. When a recipient becomes primary for a kidney offer, the patient/center must decide to accept or reject that offer without knowing what may come next and when that offer may come. The result is serial one donor one recipient accept/reject decision making. Offers can be turned down based on HLA matching or other match characteristics (donor size, age, Cr, KDPI, etc.), but the recipient doesn't know when the next offer will come or if it will be better than the current one.

In contrast, organ cryopreservation could allow for banking multiple organs (potentially 100's or 1000's), and then the center/patient can decide which organ among the many to select. Considerations such as HLA match and donor quality can be made on all of the organs in the bank. Such a practice is similar to the living donor transplants made possible by the paired-exchange program. In that program, patients have the potential to select from

several donors at once, albeit not as many as could be achieved with an organ bank.

Of course, this would be disruptive to the organ allocation process. Who gets to pick, and when do they get to choose? Are they limited to specific donors, such as the current low, medium, or high KDPI limits for allocation? Which organs get banked, and which get transplanted fresh? These and many other questions remain to be answered. One possible path would be to split some percent of organs off for banking while keeping some for direct use. There would undoubtedly be debate and policy considerations about how to “fund” the bank initially so patients could choose from multiple potential donors.

Further, clinical translation will bring several ethical, legal, and social implications (ELSI). We will need to define how cryopreserved organs function compared to other preservation technologies. These data will help determine the best manner for clinical allocation or answer the question, “who gets what organ?” We have already begun the discussion of the ELSI considerations and have included experts in bioethics and law (Susan Wolf) and organ allocation policy (Tim Pruett), both of whom contributed to this manuscript.

- Enabling tolerance induction protocols. Several promising tolerance protocols require days, weeks, or months of patient or immune cell preparation before transplant. For example, the current MGH, Stanford, and Northwestern combined hematopoietic stem cell/kidney transplants require time for conditioning the patient (irradiation, chemotherapy) and preparing hematopoietic cells. Current clinical trials are ongoing, but for the Stanford and Northwestern protocols, the stem cells (and other cell populations) require several weeks to prepare and conduct QC/QA testing. Other protocols, such as coupled cell tolerance, require at least seven days of patient conditioning before transplant. Each of these protocols will require more time for patient conditioning than enabled by normothermic perfusion (1-3 days), thus limiting many trials to living donor transplants only. Organ cryopreservation could facilitate tolerance trials for deceased donor transplants as well.
- Increasing organ utilization and the number of organs available for transplant. We agree that most high-quality organs are used directly for kidney transplant. However, as many as 20% of organs recovered for transplant are ultimately discarded. Some of these might be used in selected recipients, even if they are from less-than-optimal donors. All reasonable organs could be cryopreserved. In specific patients, it may be appropriate to use one of the banked organs, even if less than ideal, given certain circumstances, such as lack of dialysis access. Estimates suggest that a 5% increase in deceased kidney donation would save \$4.7B and gain 30,870 quality-adjusted life years (Chen, *MDM Policy Pract.* 2021). A 2016 White House Organ Summit identified organ preservation as one of the key avenues to reduce the size of the organ waiting list.
- Addition to the manuscript. To clarify these issues in the manuscript, we added the following to the discussion: “... These approaches, and ours, allow for national organ sharing, more time to perform cross-matching, and converting transplants to daytime events. However, nanowarming uniquely would improve donor/recipient matching by increasing the number of organs available for consideration at one time. In contrast, other approaches continue the practice of one at a time decision-making for every potential organ offer. Further, nanowarming would allow better patient preparation before surgery, enable tolerance protocols that take days to weeks of conditioning or preparation, and increase organ utilization by banking organs even if an appropriate recipient has not yet been identified.”

Conclusion: While we agree with the reviewers that there is still much to be learned, this study is a promising indicator that vitrification/nanowarming technology may one day help achieve clinical organ banking.

REVIEWERS' COMMENTS

Reviewer #1 (Remarks to the Author):

I see that all points have been covered. I have nothing else to add

Reviewer #2 (Remarks to the Author):

I appreciate the author's completeness in their responses and the modifications they have made based on my suggestions. I wish to congratulate the author's again for this very powerful and exciting accomplishment that has potential for high impact in the transplant field.

Response to reviewers

Reviewer 1: "I see that all points have been covered. I have nothing else to add."

Response: We thank the reviewer for their helpful comments and careful review.

Reviewer 2: "I appreciate the author's completeness in their responses and the modifications they have made based on my suggestions. I wish to congratulate the authors again for this very powerful and exciting accomplishment that has potential for high impact in the transplant field."

Response: We thank the reviewer for their helpful comments and careful review.